# SpikeGrad: An ANN-equivalent Computation Model for Implementing Backpropagation with Spikes

**Johannes C. Thiele, Olivier Bichler & Antoine Dupret**
CEA, LIST
91191 Gif-sur-Yvette, France
`{johannes.thiele,olivier.bichler,antoine.dupret}@cea.fr`

## Abstract

Event-based neuromorphic systems promise to reduce the energy consumption of deep neural networks by replacing expensive floating point operations on dense matrices by low energy, sparse operations on spike events. While these systems can be trained increasingly well using approximations of the backpropagation algorithm, this usually requires high precision errors and is therefore incompatible with the typical communication infrastructure of neuromorphic circuits. In this work, we analyze how the gradient can be discretized into spike events when training a spiking neural network. To accelerate our simulation, we show that using a special implementation of the integrate-and-fire neuron allows us to describe the accumulated activations and errors of the spiking neural network in terms of an equivalent artificial neural network, allowing us to largely speed up training compared to an explicit simulation of all spike events. This way we are able to demonstrate that even for deep networks, the gradients can be discretized sufficiently well with spikes if the gradient is properly rescaled. This form of spike-based backpropagation enables us to achieve equivalent or better accuracies on the MNIST and CIFAR10 datasets than comparable state-of-the-art spiking neural networks trained with full precision gradients. The algorithm, which we call *SpikeGrad*, is based on accumulation and comparison operations and can naturally exploit sparsity in the gradient computation, which makes it an interesting choice for a spiking neuromorphic systems with on-chip learning capacities.

## 1 Introduction

Spiking neural networks (SNNs) are a new generation of artificial neural network models (Maass, 1997) that try to harness properties of biological neurons to build energy efficient spiking neuromorphic systems. Processing in traditional artificial neural networks (ANNs) is based on parallel processing of operations on dense tensors of fixed length. In contrast to this, spiking neuromorphic systems communicate with asynchronous events, which allows dynamic, data dependent computation that can exploit high temporal and spatial sparsity.

The recent years have seen a large number of approaches devoted to optimization of spiking neural networks with the backpropagation algorithm, either by converting ANNs to SNNs (Diehl et al., 2015; Esser et al., 2016; Rueckauer et al., 2017; Sengupta et al., 2019) or by simulating spikes explicitly in the forward pass and optimizing these dynamics with floating point gradients (Lee et al., 2016; Yin et al., 2017; Wu et al., 2018b;c; Severa et al., 2019; Jin et al., 2018; Bellec et al., 2018; Zenke & Ganguli, 2018; Shrestha & Orchard, 2018). These methods aim to optimize SNNs for efficient inference, and backpropagation is performed offline on a standard computing system. It would however be desirable to also enable on-chip learning in neuromorphic chips using the power of the backpropagation algorithm, and to maintain the advantages of spike-based processing also in the error propagation phase.

Previous work on the implementation of backpropagation with spikes is mostly concerned with biological plausibility. A non-spiking version of biologically inspired backpropagation is presented

by Sacramento et al. (2018). Guerguiev et al. (2017), Neftci et al. (2017) and Samadi et al. (2017) introduce spike-based versions of the backpropagation algorithm using variants of (direct) feedback alignment (Lillicrap et al., 2016; Nøkland, 2016). The exact backpropagation algorithm, which backpropagates through symmetric weights, might however be required to achieve good inference performance on large-scale deep neural networks (Baldi & Sadowski, 2016; Bartunov et al., 2018). O'Connor & Welling (2016) and Thiele et al. (2019) present implementations of standard backpropagation where the gradient is coded into spikes and propagated through symmetric weights. In the same spirit, our work is mostly concerned with exploiting spike based information encoding for energy efficient processing, which means that inference performance and operational simplicity will be preferred over biological plausibility and complex neuron models.

We demonstrate how backpropagation can be seamlessly integrated into the spiking neural network framework by using a second accumulation compartment that discretizes the error into spikes. By additionally weighting the activity counters by the learning rate, we obtain a system that is able to perform learning and inference based on accumulations and comparisons alone. As for the forward pass, this allows us to use the dynamic precision computation provided by the discretization of all operations into spike events, and to exploit the sparsity of the gradient. Using a similar reasoning as Binas et al. (2016) and Wu et al. (2019) have applied to forward propagation in SNNs, we show that the system obtained in this way can be mapped to an integer activation ANN whose activations are equivalent to the accumulated neuron responses for both the forward and the backward propagation phase. This allows us to simulate training of large-scale SNNs efficiently on graphics processing units (GPUs), using their equivalent ANN. Additionally, in contrast to conversion methods that approximate pre-trained ANNs with SNNs, this method guarantees that the inference precision of the SNN will be equivalent to the ANN. In contrast to O'Connor & Welling (2016), this is true for any number of spikes and arbitrary spike order. We demonstrate classification accuracies equivalent or superior to existing implementations of SNNs trained with full precision gradients, and comparable to the precision of standard ANNs using similar topologies. This is the first time competitive classification performances are reported on the CIFAR10 and CIFAR100 datasets using a large-scale SNN where both training and inference are fully implemented with spikes. To the best of our knowledge, our work provides for the first time a demonstration of how the sparsity of the gradient during backpropagation could be exploited within a large-scale SNN processing structure.

## 2 THE *SpikeGrad* ALGORITHM

We begin with the description of *SpikeGrad*, the spike-based backpropagation algorithm. The algorithm mainly consists of special implementations of the integrate-and-fire (IF) neuron model, that are used to encode and propagate information in the forward and backward pass.

For each training example/mini-batch, integration is performed from $t = 0$ to $t = T$ for the forward pass and from $t = T + \Delta t$ to $t = \mathcal{T}$ in the backward pass. Since no explicit time is used in the algorithm, $\Delta t$ represents symbolically the (very short) time between the arrival of an incoming spike and the response of the neuron, which is only used here to describe causality.

**Spike discretization of the forward pass** For forward propagation, the architecture is described by multiple layers (labeled by $l \in [0, L]$) of IF neurons with integration variable $V_i^l(t)$ and threshold $\Theta_{\text{ff}}$:

$$V_i^l(t + \Delta t) = V_i^l(t) - \Theta_{\text{ff}} s_i^l(t) + \sum_j w_{ij}^l s_j^{l-1}(t), \quad V_i^l(0) = b_i^l. \tag{1}$$

The variable $w_{ij}^l$ is the weight and $b_i^l$ a bias value. The spike activation function $s_i^l(t) \in \{-1, 0, 1\}$ is a function which triggers a signed spike event depending on the internal variables of the neuron. It will be shown later that the specific choice of the activation function is fundamental for the mapping to an equivalent ANN. After a neuron has fired, its integration variable is decremented or incremented by the threshold value $\Theta_{\text{ff}}$, which is represented by the second term on the r.h.s. of equation 1.

As a representation of the neuron activity, we use a trace $x_i^l(t)$ which accumulates spike information over a single example:

$$x_i^l(t + \Delta t) = x_i^l(t) + \eta s_i^l(t). \tag{2}$$

By weighting the activity with the learning rate $\eta$ we avoid performing a multiplication when weighting the input with the learning rate for the weight update equation 8.

**Implementation of implicit ReLU and surrogate activation function derivative**   It is possible to define an implicit activation function based on how the neuron variables affect the spike activation function $s_i^l(t)$. In our implementation, we use the following fully symmetric function to represent linear activation functions (used for instance in pooling layers):

$$s_i^{l,\mathrm{lin}}\left(V_i^l(t)\right) := \begin{cases} 1 & \text{if } V_i^l(t) \geq \Theta_{\mathrm{ff}} \\ -1 & \text{if } V_i^l(t) \leq -\Theta_{\mathrm{ff}} \\ 0 & \text{otherwise} \end{cases}. \tag{3}$$

The following function corresponds to the rectified linear unit (ReLU) activation function:

$$s_i^{l,\mathrm{ReLU}}\left(V_i^l(t), x_i^l(t)\right) := \begin{cases} 1 & \text{if } V_i^l(t) \geq \Theta_{\mathrm{ff}} \\ -1 & \text{if } V_i^l(t) \leq -\Theta_{\mathrm{ff}} \text{ and } x_i^l(t) > 0 \\ 0 & \text{otherwise} \end{cases}. \tag{4}$$

The pseudo-derivative of the activation function is denoted symbolically by $S_i'^l$. We use $S_i'^{l,\mathrm{lin}}(T) = 1$ for the linear case. For the ReLU, we use a surrogate of the form:

$$S_i'^{l,\mathrm{ReLU}}(T) := \begin{cases} 1 & \text{if } V_i^l(T) > 0 \text{ or } x_i^l(T) > 0 \\ 0 & \text{otherwise} \end{cases}. \tag{5}$$

These choices will be motivated in the following sections. Note that the derivatives depend only on the final states of the neurons at time $T$.

**Discretization of the error into spikes**   For gradient backpropagation, we introduce a second compartment with threshold $\Theta_{\mathrm{bp}}$ in each neuron, which integrates error signals from higher layers. The process discretizes errors in the same fashion as the forward pass discretizes an input signal into a sequence of signed spike signals:

$$U_i^l(t + \Delta t) = U_i^l(t) - \Theta_{\mathrm{bp}} z_i^l(t) + \sum_k w_{ki}^{l+1} \delta_k^{l+1}(t). \tag{6}$$

To this end, we introduce a ternary *error spike activation function* $z_i^l(t) \in \{-1, 0, 1\}$ which is defined in analogy to equation 3 using the error integration variable $U_i^l(t)$ and the backpropagation threshold $\Theta_{\mathrm{bp}}$. The error is then obtained by gating this ternarized variable $z_i^l(t)$ with one of the surrogate activation function derivatives of the previous section (linear or ReLU):

$$\delta_i^l(t) = z_i^l(t) S_i'^l(T). \tag{7}$$

This ternary spike signal is backpropagated through the weights to the lower layers and also applied in the update rule of the weight increment accumulator $\omega_{ij}^l$:

$$\omega_{ij}^l(t + \Delta t) = \omega_{ij}^l(t) - \delta_i^l(t) x_j^{l-1}(T), \tag{8}$$

which is triggered every time an error spike (equation 7) is backpropagated. The weight updates are accumulated during error propagation and are applied after propagation is finished to update each weight simultaneously. In this way, the backpropagation of errors and the weight update will, exactly as forward propagation, only involve additions and comparisons of floating point numbers.

The *SpikeGrad* algorithm can also be expressed in an event-based formulation, described in algorithms 1, 2 and 3. This formulation is closer to how the algorithm would be implemented in an actual SNN hardware implementation of the IF firing dynamics.

**Loss function and error scale**   We use the cross entropy loss function in the final layer applied to the softmax of the total integration $V_i^L(T)$ (no spikes are triggered in the top layer during inference). This requires more complex operations than accumulations, but is negligible if the number of classes is small. To make sure that sufficient error spikes are triggered in the top layer, and that error spikes arrive even in the lowest layer of the network, we apply a scaling factor $\alpha$ to the error values before transferring them to $U_i^L$. This scaling factor also implicitly sets the precision of the gradient, since a higher number of spikes means that a large range of values can be represented. To counteract the relative increase of the gradient scale, the learning rates have to be rescaled by a factor $1/\alpha$.

**Input encoding**   As pointed out in Rueckauer et al. (2017) and Wu et al. (2018c), it is crucial to maintain the full precision of the input image to obtain good performances on complex standard benchmarks with SNNs. One possibility is to encode the input in a large number of spikes (Sengupta et al., 2019). Another possibility, which has been shown to require a much lower number of spikes in the network, is to multiply the input values directly with the weights of the first layer (just like in a standard ANN). The drawback is that the first layer then requires multiplication operations. The additional cost of this procedure may however be negligible if all other layers can profit from spike-based computation. This problematic does not exist for stimuli which are natively encoded in spikes.

---

**Algorithm 1** Forward

---

**function** PROPAGATE($[l, i, j], s$)
    $V_i^l \leftarrow V_i^l + s \cdot w_{ij}^l$
    $s_i^l \leftarrow s_i^l(V_i^l, x_i^l)$ ▷ spike activation function
    **if** $s_i^l \neq 0$ **then**
        $V_i^l \leftarrow V_i^l - s_i^l \cdot \Theta_{\text{ff}}$
        $x_i^l \leftarrow x_i^l + \eta s_i^l$
        **for** $k$ in $l + 1$ connected to $i$ **do**
            PROPAGATE($[l + 1, k, i], s_i^l$)

---

**Algorithm 2** Backward

---

**function** BACKPROPAGATE($[l, i, k], \delta$)
    $U_i^l \leftarrow U_i^l + \delta \cdot w_{ki}^{l+1}$
    $z_i^l \leftarrow z_i^l(U_i^l)$     ▷ error activation function
    $\delta_i^l \leftarrow z_i^l \cdot S_i'^l$
    **if** $z_i^l \neq 0$ **then**
        $U_i^l \leftarrow U_i^l - z_i^l \cdot \Theta_{\text{bp}}$
        **for** $j$ in layer $l - 1$ connected to $i$ **do**
            BACKPROPAGATE($[l - 1, j, i], \delta_i^l$)
            $\omega_{ij}^l \leftarrow \omega_{ij}^l - \delta_i^l \cdot x_j^{l-1}$

---

**Algorithm 3** Training of single example/batch

---

**init:** $V \leftarrow b, U \leftarrow 0, x \leftarrow 0, \omega \leftarrow 0$    ▷ variables in **bold** describe all neurons in network/layer
**while** input spikes $s_i^{in}$ **do**
    **for** $k$ in $l = 0$ receiving $s_i^{in}$ **do**                          ▷ spikes corresponding to training input
        PROPAGATE($[0, k, i], s_i^{in}$)
$S' \leftarrow S'(V, x)$                                            ▷ calculate surrogate derivatives
$U^L \leftarrow \alpha \cdot \partial \mathcal{L} / \partial V^L$                        ▷ calculate classfication error
**for** $i$ in $l = L$ **do**
    **while** $|U_i^L| \geq \Theta_{\text{bp}}$ **do**                              ▷ backpropagate error spikes
        BACKPROPAGATE($[L, i, -], 0$)              ▷ last layer receives no error
$w \leftarrow w + \omega$                       ▷ update weights with weight update accumulator

---

## 3   FORMULATION OF THE EQUIVALENT ANN

The simulation of the temporal dynamics of spikes in *SpikeGrad* requires a large number of time steps or events if activation or error values are large. It would therefore be extremely beneficial if we were able to map the SNN to an equivalent ANN that can be trained much faster on standard hardware. In this section, we demonstrate that it is possible to find such an ANN using the forward and backward propagation dynamics of the *SpikeGrad* IF neuron model described in the previous section.

**Spike discretization error**   We start our analysis with equation 1. We reorder the terms and sum over the increments $\Delta V_i^l(t) = V_i^l(t + \Delta t) - V_i^l(t)$ every time the integration variable is changed either by a spike that arrives at time $t_j^s \in [0, T]$ via connection $j$, or by a spike that is triggered at time $t_i^s \in [0, T]$. With the initial conditions $V_i^l(0) = b_i^l, s_i^l(0) = 0$, we obtain the final value $V_i^l(T)$:

$$V_i^l(T) = \sum_{t_j^s, t_i^s} \Delta V_i^l = -\Theta_{\text{ff}} \sum_{t_i^s} s_i^l(t_i^s) + \sum_j w_{ij}^l \sum_{t_j^s} s_j^{l-1}(t_j^s) + b_i^l \tag{9}$$

By defining the total transmitted output of a neuron as $S_i^l := \sum_{t_i^s} s_i^l(t_i^s)$ we obtain:

$$\frac{1}{\Theta_{\text{ff}}} V_i^l(T) = \mathbb{S}_i^l - S_i^l, \quad \mathbb{S}_i^l := \frac{1}{\Theta_{\text{ff}}} \left( \sum_j w_{ij}^l S_j^{l-1} + b_i^l \right) \tag{10}$$

The same reasoning can be applied to backpropagation of the gradient. We define the summed responses over error spikes times $\tau_j^s \in [T + \Delta t, \mathcal{T}]$ as $Z_i^l := \sum_{\tau_i^s} z_i^l(\tau_i^s)$ to obtain:

$$\frac{1}{\Theta_{bp}} U_i^l(\mathcal{T}) = \mathbb{Z}_i^l - Z_i^l, \quad \mathbb{Z}_i^l := \frac{1}{\Theta_{bp}} \left( \sum_k w_{ki}^{l+1} E_k^{l+1} \right) \tag{11}$$

$$E_k^{l+1} = \sum_{\tau_k^s} \delta_k^{l+1}(\tau_k^s) = \sum_{\tau_k^s} S_k'^{l+1}(T) z_k^{l+1}(\tau_k^s) = S_k'^{l+1}(T) Z_k^{l+1}. \tag{12}$$

In both equation 10 and equation 11, the terms $\mathbb{S}_i^l$ and $\mathbb{Z}_i^l$ are equivalent to the output of an ANN with signed integer inputs $S_j^{l-1}$ and $E_k^{l+1}$. $1/\Theta_{ff}$ and $1/\Theta_{bp}$ are implicit scaling factors of activations and gradients. If gradients shall not be explicitly rescaled, backpropagation requires $\Theta_{bp} = \Theta_{ff}$. The values of the residual integrations $1/\Theta_{ff} V_i^l(T)$ and $1/\Theta_{bp} U_i^l(\mathcal{T})$ therefore represent the *spike discretization error* $\mathrm{SDE}_{ff} := \mathbb{S}_i^l - S_i^l$ or $\mathrm{SDE}_{bp} := \mathbb{Z}_i^l - Z_i^l$ between the ANN outputs $\mathbb{S}_i^l$ and $\mathbb{Z}_i^l$ and the accumulated SNN outputs $S_i^l$ and $Z_i^l$. Since we know that $V_i^l(T) \in (-\Theta_{ff}, \Theta_{ff})$ and $U_i^l(\mathcal{T}) \in (-\Theta_{bp}, \Theta_{bp})$, this gives bounds of $|\mathrm{SDE}_{ff}| < 1$ and $|\mathrm{SDE}_{bp}| < 1$.

So far we can only represent linear functions. We now consider an implementation where the ANN applies a ReLU activation function instead. The SDE in this case is:

$$\mathrm{SDE}_{ff}^{\mathrm{ReLU}} := \mathrm{ReLU}\left(\mathbb{S}_i^l\right) - S_i^l. \tag{13}$$

We can calculate the error by considering that equation 4 forces the neuron in one of two regimes (note that $x_i^l > 0 \Leftrightarrow S_i^l > 0$): In one case, $S_i^l = 0$, $V_i^l(T) < \Theta_{ff}$ (this includes $V_i^l(T) \leq -\Theta_{ff}$). This implies $\mathbb{S}_i^l = 1/\Theta_{ff} V_i^l(T)$ and therefore $|\mathrm{SDE}_{ff}^{\mathrm{ReLU}}| < 1$ (or even $|\mathrm{SDE}_{ff}^{\mathrm{ReLU}}| = 0$ if $V_i^l(T) \leq 0$). In the other case, $S_i^l > 0$, $V_i^l(t) \in (-\Theta_{ff}, \Theta_{ff})$, where equation 4 is equivalent to equation 3.

This equivalence motivates the choice of equation 5 as a surrogate derivative for the SNN: the condition $(V_i^l(T) > 0$ or $x_i^l(T) > 0)$ can be seen to be equivalent to $\mathbb{S}_i^l(T) > 0$, which defines the derivative of a ReLU. Finally, for the total weight increment $\Delta w_{ij}^l$, it can be seen from equation 2 and equation 8 that:

$$x_i^l(T) = \sum_{t_i^s} \Delta x_i^l(t_i^s) = \eta S_i^l, \quad \Rightarrow \quad \Delta w_{ij}^l(\mathcal{T}) = \sum_{\tau_i^s} \Delta \omega_{ij}^l(\tau_i^s) = -\eta S_j^{l-1} E_i^l, \tag{14}$$

which is exactly the weight update formula of an ANN defined on the accumulated variables. We have therefore demonstrated that the SNN can be represented by an ANN by replacing all $S$ and $Z$ by $\mathbb{S}$ and $\mathbb{Z}$ and applying the corresponding activation function directly on these variables. The error that will be caused by this substitution compared to using the accumulated variables $S$ and $Z$ of an SNN is described by the SDE. This ANN can now be used for training of the SNN on GPUs. The *SpikeGrad* algorithm formulated on the variables $s$, $z$, $\delta$ and $x$ represents the algorithm that would be implemented on a dedicated, event-based *spiking* neural network hardware implementation of the IF neurons. We will now demonstrate how the SDE can be further reduced to obtain an ANN and SNN that are exactly equivalent.

**Response equivalence**   For a large number of spikes, the SDE may be negligible compared to the activation of the ANN. However, in a framework whose objective it is to minimize the number of spikes emitted by each neuron, this error can have a potentially large impact.

One option to reduce the error between the ANN and the SNN output is to constrain the ANN during training to integer values. One possibility is to round the ANN outputs:

$$\mathbb{S}_i^{l,\mathrm{round}} := \mathrm{round}[\mathbb{S}_i^l] = \mathrm{round}\left[ \frac{1}{\Theta_{ff}} \left( \sum_j w_{ij}^l S_j^{l-1} + b_i^l \right) \right], \tag{15}$$

The round function here rounds to the next integer value, with boundary cases rounded *away* from zero. This behavior can be implemented in the SNN by a modified spike activation function which is applied after the full stimulus has been propagated. To obtain the exact response as the ANN, we have to take into account the current value of $S_i^l$ and modify the threshold values:

$$s_i^{l,\mathrm{res}}\left(V_i^l(T), S_i^l\right) := \begin{cases} 1 & \text{if } V_i^l(T) > \Theta_{ff}/2 \text{ or } (S_i^l \geq 0, \ V_i^l(T) = \Theta_{ff}/2) \\ -1 & \text{if } V_i^l(T) < -\Theta_{ff}/2 \text{ or } (S_i^l \leq 0, \ V_i^l(T) = -\Theta_{ff}/2) \\ 0 & \text{otherwise} \end{cases} \tag{16}$$

Because this spike activation function is applied only to the residual values, we call it the *residual spike activation function*. The function is applied to a layer after all spikes have been propagated with the standard spike activation function (equation 3 or equation 4). We start with the lowest layer and propagate all residual spikes to the higher layers, which use the standard activation function. We then proceed with setting the next layer to residual mode and propagate the residual spikes. This is continued until we arrive at the last layer of the network.

By considering all possible rounding scenarios, it can be seen that equation 16 indeed implies:

$$S_i^l + s_i^{l,\text{res}}\left(V_i^l(T), S_i^l\right) = \text{round}[S_i^l + 1/\Theta_{\text{ff}} V_i^l(T)] = \text{round}[\mathbb{S}_i^l]. \tag{17}$$

The same principle can be applied to obtain integer-rounded error propagation:

$$\mathbb{Z}_i^{l,\text{round}} := \text{round}\left[\mathbb{Z}_i^l\right] = \text{round}\left[\frac{1}{\Theta_{\text{bp}}}\left(\sum_k w_{ki}^{l+1} E_k^{l+1}\right)\right]. \tag{18}$$

We have to apply the following modified spike activation function in the SNN after the full error has been propagated by the standard error spike activation function:

$$z_i^{l,\text{res}}\left(U_i^l(\mathcal{T}, Z_i^l)\right) := \begin{cases} 1 & \text{if } U_i^l(\mathcal{T})) > \Theta_{\text{bp}}/2 \text{ or } (Z_i^l \geq 0,\ U_i^l(\mathcal{T}) = \Theta_{\text{bp}}/2) \\ -1 & \text{if } U_i^l(\mathcal{T}) < -\Theta_{\text{bp}}/2 \text{ or } (Z_i^l \leq 0,\ U_i^l(\mathcal{T}) = -\Theta_{\text{bp}}/2) , \\ 0 & \text{otherwise} \end{cases} \tag{19}$$

which implies:

$$Z_i^l + z_i^{l,\text{res}}\left(U_i^l(\mathcal{T}), Z_i^l\right) = \text{round}[Z_i^l + 1/\Theta_{\text{bp}} U_i^l(\mathcal{T})] = \text{round}[\mathbb{Z}_i^l]. \tag{20}$$

We have therefore shown that the SNN will after each propagation phase have exactly the same accumulated responses as the corresponding integer activation ANN. The same principle can be applied to obtain other forms of rounding (e.g. floor and ceil), if equation 16 and equation 19 are modified accordingly.

**Computational complexity estimation**    Note that we have only demonstrated the equivalence of the accumulated neuron responses. However, for each of the response values, there is a large number of possible combinations of 1 and $-1$ values that lead to the same response. The computational complexity of the event-based algorithm depends therefore on the total number $n$ of these events. The best possible case is when the accumulated response value $S_i^l$ is represented by exactly $|S_i^l|$ spikes. In the worst case, a large number of additional redundant spikes is emitted which sum up to 0. The maximal number of spikes in each layer is bounded by the largest possible integration value that can be obtained. This depends on the maximal absolute weight value $w_{\text{max}}^l$, the number of connections $N_{\text{in}}^l$ and the number of spike events $n^{l-1}$ each connection receives, which is given by the maximal value of the previous layer (or the input in the first layer):

$$n_{\text{min}}^l = |S_i^l|, \quad n_{\text{max}}^l = \left\lfloor \frac{1}{\Theta_{\text{ff}}} N_{\text{in}}^l w_{\text{max}}^l n_{\text{max}}^{l-1} \right\rfloor. \tag{21}$$

The same reasoning applies to backpropagation. Our experiments show that for input encodings where the input is provided in a continuous fashion, and weight values that are much smaller than the threshold value, the deviation from the best case scenario is rather small. This is because in this case the sub-threshold integration allows to average out the fluctuations in the signal. This way the firing rate stays rather close to its long term average and few redundant spikes are emitted. For the total number of spikes $n$ in the full network on the CIFAR10 test set, we obtain empirically $n-n_{\text{min}}/n_{\text{min}} < 0.035$.

## 4    EXPERIMENTS

**Classification performance**    Tables 1 and 2 compare the state-of-the-art results for SNNs on the MNIST and CIFAR10 datasets. It can be seen that in both cases, our results are competitive with respect to the state-of-the-art results of other SNNs trained with high precision gradients. Compared to results using the same topology, our algorithm performs at least equivalently.

Table 1: Comparison of different state-of-the-art spiking CNN architectures on MNIST. * indicates that the same topology (28x28-15C5-P2-40C5-P2-300-10) was used.

| Architecture | Method | Rec. Rate (max[mean±std]) |
|---|---|---|
| Wu et al. (2018b)* | BP float gradient | 99.42% |
| Rueckauer et al. (2017) | CNN converted to SNN | 99.44% |
| Jin et al. (2018)* | BP float gradient | 99.49% |
| **This work*** | BP float gradient | **99.48**[99.36 ± 0.06]% |
| **This work*** | BP spike gradient | **99.52**[99.38 ± 0.06]% |

Table 2: Comparison of different state-of-the-art spiking CNN architectures on CIFAR10. * indicates that the same topology (32x32-128C3-256C3-P2-512C3-P2-1024C3-512C3-1024-512-10) was used.

| Architecture | Method | Rec. Rate (max[mean±std]) |
|---|---|---|
| Rueckauer et al. (2017) | CNN converted SNN (with BN) | 90.85% |
| Sengupta et al. (2019) | VGG-16 converted to SNN | 91.55% |
| Wu et al. (2018c)* | BP float gradient (no NeuNorm) | 89.32% |
| **This work*** | BP float gradient | **89.72**[89.38 ± 0.25]% |
| **This work*** | BP spike gradient | **89.99**[89.49 ± 0.28]% |

The final classification performance of the network as a function of the error scaling term $\alpha$ in the final layer can be seen in figure 1. Previous work on low bitwidth gradients (Zhou et al., 2018) found that gradients usually require a higher precision than both weights and activations. Our results also indicate that a certain minimum number of error spikes is necessary to achieve convergence. This strongly depends on the depth of the network and if enough spikes are triggered to provide sufficient gradient signal in the bottom layers. For the CIFAR10 network, convergence becomes unstable for approximately $\alpha < 300$. If the number of operations is large enough for convergence, the required precision for the gradient does not seem to be extremely high. On the MNIST task, the difference in test performance between a gradient rescaled by a factor of 50 and a gradient rescaled by a factor of 100 becomes insignificant. In the CIFAR10 task, this is true for a rescaling by 400 or 500. Also the results obtained with the float precision gradients in tables 1 and 2 demonstrate the same performance, given the range of the error.

To investigate the performance of *SpikeGrad* on databases with a larger number of classes, we also ran an additional experiment on the CIFAR100 dataset. Using the exactly same architecture as for CIFAR10 with $\alpha = 500$ (besides the final layer that is increased to 100 classes), we obtain a maximal classification score of $64.40\%$. Running the same experiment with only the forward pass encoded in spikes, but floating point gradients, we obtain $64.69\%$. This result could probably be improved by adapting the architecture better to the dataset. However, it demonstrates that coding the gradient into spikes does not lead to a large precision loss even in this scenario.

**Sparsity in backpropagated gradient**  To evaluate the potential efficiency of the spike coding scheme relative to an ANN, we use the metric of relative synaptic operations. A synaptic operation corresponds to a multiply-accumulate (MAC) in the case of an ANN, and a simple accumulation (ACC) in the case of an SNN. This metric allows us to compare networks based on their fundamental operation. The advantage of this metric is the fact that it does not depend on the exact implementation of the operations (for instance the number of bits used to represent each number). Since an ACC is however generally cheaper and easier to implement than a MAC, we can be sure that an SNN is more efficient in terms of its operations than the corresponding ANN if the number of ACCs is smaller than the number of MACs (without considering potential hardware overheads of event-based computing).

Numbers were obtained with the integer activations of the equivalent ANN to keep simulation times tractable. As previously explained, the integer response of the equivalent ANN represents the best case scenario for the SNN, i.e. the activation encoding with the lowest number of spikes. The actual

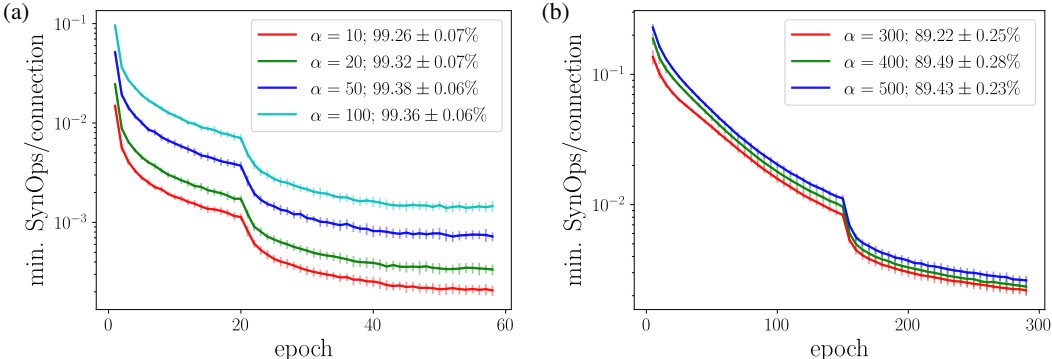

Figure 1: Number of relative synaptic operations during backpropagation for different error scaling factors $\alpha$ as a function of the epoch. Numbers are based on activation values of the equivalent integer activation ANN. Test performance with error is given for each $\alpha$. (a) MNIST. (b) CIFAR10.

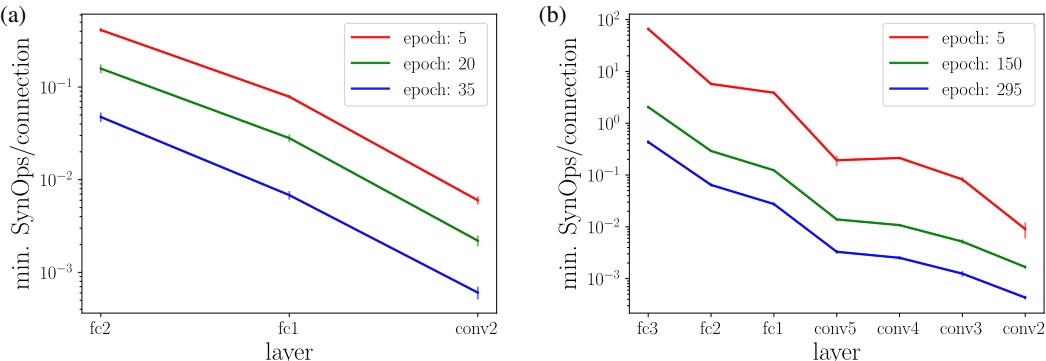

Figure 2: Number of relative synaptic operations during backpropagation in each layer (connections in direction of backpropagation) for different epochs. Numbers are based on activation values of the equivalent integer activation ANN. (a) MNIST with $\alpha = 100$. (b) CIFAR10 with $\alpha = 500$.

number of events and synaptic operations in an SNN may therefore slightly deviate from these numbers.

In figure 1, it can be seen that the number of operations decreases with increasing inference precision of the network. This is a result of the decrease of error in the classification layer, which leads to the emission of a smaller number of error spikes. Figure 2 demonstrates how the number of operations during the backpropagation phase is distributed in the layers of the network (the float input bottom layer and average pooling layers were omitted). While propagating deeper down the network, the relative number of operations decreases and the error becomes increasingly sparse. This tendency is consistent during the whole training process for different epochs.

## 5 DISCUSSION AND CONCLUSION

Using spike-based propagation of the gradient, we demonstrated that the paradigm of event-based information propagation can be easily translated to the backpropagation algorithm. We have not only shown that competitive inference performance can be achieved, but also that gradient propagation seems particularly suitable to leverage spike-based processing by exploiting high signal sparsity. For both forward and backward propagation, *SpikeGrad* requires a similar event-based communication infrastructure between neurons, which simplifies a possible spiking hardware implementation. One restriction of our algorithm is the need for negative spikes, which could be problematic for some neuromorphic hardware platforms.

In particular the topology used for CIFAR10 classification is rather large for the given task. We decided to use the same topologies as other state-of-the-art SNNs to allow for better comparison. The relatively large number of parameters of these topologies may to a certain extent explain the very low number of relative synaptic operations we observe during backpropagation. It has been demonstrated that the number of parameters in many ANNs can be strongly reduced without a significant impact on performance (Han et al., 2016). It remains to show that the spike coding approach provides sufficient gains also in topologies that are optimized for minimal computation and memory requirements. This seems at least possible, since the reduction of operation count in SNNs is mainly caused by two factors: the first factor is actual sparsity, that means the number of zeros that are produced by thresholding and the ReLU activation function. The second factor is the more fine grained distribution of activation and error values. If most values are represented by small integers, only a few spikes are necessary to encode them. The consequence is that even topologies that are compressed and therefore have lower sparsity may be able to profit from spike coding if most of the activation or error values are small. This is typically the case for networks quantized to low precision integer values (see for instance (Zhou et al., 2018) or Wu et al. (2018a)) and could lead to interesting synergies between our spike-based computation model and ANN quantization methods.

Our analysis of the potential energy efficiency of *SpikeGrad* in a dedicated SNN architecture is based on counting the number of basic operations that are performed at the synapses of a neuron. This does not take into account the potential hardware overhead that is introduced by the implementation of the IF neuron model and the event-based processing scheme. If the low operation count can in the end be translated into higher processing speed or increased energy efficiency, compared to a normal ANN, depends on the particular SNN hardware implementation. Such dedicated neuromorphic hardware is the subject of current research efforts (see for instance Merolla et al. (2014) and Davies et al. (2018)). We assumed in this work that such a system might be built and used operation count as our principal metric, since it enables us to easily approximate the energy consumption of a spiking neuromorphic chip if the average cost of such an operation is known. Our work shows that a hardware implementation of *SpikeGrad* might profit sufficiently from the low operation count that energy or speed improvements could be possible at least in certain cases.

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

# A    APPENDIX

# B    ADDITIONAL EXPLANATIONS

## B.1    ROUND FUNCTION

It can be seen that the residual spike activation function covers indeed all possible cases. In particular, the boundary cases are rounded correctly: for $V_i = \Theta_{\text{ff}}$, we obtain $\mathbb{S} = S_i + 0.5$. For $S_i \geq 0$, this should be rounded to $\text{round}[\mathbb{S}] = S_i + 1$ and for $S_i < 0$, we should obtain $\text{round}[\mathbb{S}] = S_i$. Similarly,

for $V_i = -\Theta_{\text{ff}}$, we obtain $\mathbb{S} = S_i - 0.5$. For $S_i \leq 0$, this should be rounded to $\text{round}[\mathbb{S}] = S_i - 1$ and for $S_i > 0$, we should obtain $\text{round}[\mathbb{S}] = S_i$.

The same reasoning applies to error propagation.

## C  ADDITIONAL ARCHITECTURE DETAILS

### C.1  AVERAGE POOLING

All pooling layers are average pooling layers, since these are much easier to implement in a spiking neural network. Average pooling can simply be implemented as an IF neuron with constant weights:

$$w_{ij}^l = \frac{1}{p_i^l}, \tag{22}$$

where $p_i^l$ is the number of neurons in the pooling window of the neuron.

### C.2  DROPOUT

Dropout can be implemented by an additional gating variable which randomly removes some neurons during forward and backward propagation. During learning, the activations of the other neurons have to be rescaled by the inverse of the dropout probability $1/(1 - p_{\text{drop}})$. This can be implemented in the SNN framework by rescaling the threshold values by $(1 - p_{\text{drop}})$.

### C.3  MOMENTUM

While momentum is not required for the basic algorithm, we used it to speed up training. In a SNN implementation, this could be implemented by adding additional accumulators that save past gradient information.

### C.4  PARAMETER PRECISION

All real valued variables are coded with 32 bit floating point variables.

## D  EXPERIMENTS

### D.1  COMPUTING FRAMEWORK

All experiments are performed with custom CUDA/cuDNN accelerated C++ code. Training is performed on RTX 2080 Ti graphic cards.

### D.2  ERROR BARS

For all experiments, the means, errors and maximal values are calculated over 20 simulation runs.

### D.3  PREPROCESSING

No preprocessing is used on the MNIST dataset. We separate the training set of size 60000 into 50000 training and 10000 validation examples, which are used to monitor convergence. Testing is performed on the test set of 10000 examples.

For CIFAR10, the values of all color channels are divided by 255 and then rescaled by a factor of 20 to trigger sufficient activation in the network. The usual preprocessing and data augmentation is applied. For data augmentation images are padded with the image mean value by two pixels on each side and random slices of $32 \times 32$ are extracted. Additionally, images are flipped randomly along the vertical axis. We separate the training set of size 50000 into 40000 training and 10000 validation examples, which are used to monitor convergence. Testing is performed on the test set of 10000 examples.

Final scores were obtained without retraining on the validation set.

Table 3: Parameters used for training of MNIST.

| Parameter | Value |
|-----------|-------|
| Epochs | 60 |
| Batch size | 128 |
| $\Theta_{\text{ff}}$ | 1.0 |
| $\Theta_{\text{bp}}$ | 1.0 |
| Base learning rate $\eta$ | 0.1 |
| Momentum | 0.9 |
| Decay policy | mutliply by 0.1 every 20 epochs |
| Dropout (fc1 only) | 0.5 |

Table 4: Parameters used for training of CIFAR10.

| Parameter | Value |
|-----------|-------|
| Epochs | 300 |
| Batch size | 16 |
| $\Theta_{\text{ff}}$ | 1.0 |
| $\Theta_{\text{bp}}$ | 1.0 |
| Base learning rate $\eta$ | 0.001 |
| Momentum | 0.9 |
| Decay policy | multiply by 0.1 after 150 epochs |
| Dropout (all except pool and top) | 0.2 |

## D.4 HYPERPARAMETERS

The hyperparameters for training can be seen in tables 3 and 4.

The maximal inference performances in the results were achieved with $\alpha = 100$ for MNIST and $\alpha = 400$ for CIFAR10.

