# OpenReview forum: "SpikeGrad: An ANN-equivalent Computation Model for Implementing Backpropagation with Spikes"
_ICLR.cc/2020/Conference — Accept (Poster)_

### Official Review · AnonReviewer3 · 2019-10-21
**Official Blind Review #3**

**Rating:** 6

**Review:**

This paper proposes a first framework of large-scale spiking neural network that exploits the the sparsity of the gradient during backpropagation, to save training energy.
Later, it provides detailed analysis to show the equivalence of accumulated response and the corresponding integer activation ANN.

The paper is clearly written. The forward and backward process with the spike activation and error activation function respectively to save energy is clearly demonstrated. The response equivalence of the proposed architecture and integer ANNs provides theoretical gurantee for the good performance in training accuracy.

My only concern is the lack of empirical support for the energy saving of the proposal. In order to show the effectiveness of the proposal, the authors should also provide time consumptions of the SNN and normal ANN. A mere comparison on sparsity doesn't really show the advantage of the proposal, since there is some computational overhead. For a system-level improvement, it's not sufficient to show the epoch-operation relation.
If the authors could provide wall clock time comparisons, I will consider raising my score.

==========
I find the response of the authors reasonable and address some of my concerns. Therefore I'm willing to raise my score to 6.

**Experience Assessment:**

I have published in this field for several years.

**Review Assessment: Checking Correctness Of Derivations And Theory:**

I assessed the sensibility of the derivations and theory.

**Review Assessment: Checking Correctness Of Experiments:**

I assessed the sensibility of the experiments.

**Review Assessment: Thoroughness In Paper Reading:**

I read the paper at least twice and used my best judgement in assessing the paper.

---

> ### Author Response · Authors · 2019-11-08
> **Clarifications regarding the hardware efficiency and runtime of our approach**
>
> Thank you very much for your helpful review.
>
> We are happy to give our response to your main concern.
>
> SNNs require dedicated event-based digital or analog hardware implementations of the integrate-and-fire neuron, such as the Loihi chip by Intel or TrueNorth by IBM. The wall clock time of our algorithm therefore depends strongly on the exact hardware that is used to implement SpikeGrad. This paper presents algorithmic work on SNNs and we do at the moment not have access to such a specialized neuromorphic chip. The SNN in our case is simulated with a clock-based simulation of the spike dynamics, as it is common pratice in similar theoretical work on SNNs, and effectively done by the previous works that we refer to.  In general, such a simulation of an SNN on standard hardware will be much slower than that of an ANN of similar topology, since GPUs are not suitable for an efficient implementation of the event-based, sparse dynamics of SNNs.  Simulating a large SNN on standard hardware is therefore very time consuming.
> One of the main advantages of our framework is that these lengthy simulations can be avoided. We demonstrate that a special version of the integrate-and-fire neuron has equivalent accumulated responses as an integer ANN with the same weights. This means that we can be sure that an SNN that implements this integrate-and-fire neuron model in dedicated hardware will have exactly the same response as the ANN. The fact that we can simulate training of an SNN without the explicit need of a functioning SNN hardware system or a lengthy simulation is a great practical advantage of our framework. It is the main reason why our paper is the first work that is able to show that an SNN using only integrate-and-fire dynamics for BP can be trained to the same precision as SNNs that are trained using floating point gradients (floating points gradients are difficult to implement in integrate-and-fire SNN hardware).
> You are right that for a practical applications it remains to show that the integrate-and-fire dynamics can be implemented efficiently in a dedicated chip, and that the system can profit from the sparsity in computation, without too much overhead. This is indeed an ongoing question in the field of SNN hardware design, and a large number of competing approaches try to address this problem. The aim of this paper was to focus on learning algorithms that are compatible with the integrate-and-fire neuron (that is the most common SNN model), and we wanted to be agnostic with respect to the exact hardware implementation. We therefore used spike operation count as a metric, since it applies to a large number of possible SNN implementations. How efficiently these operations can be performed in a particular SNN hardware, given the additional overhead of event-based computing, is an interesting direction for future work. Your concern addresses a problem that is very relevant for the justification of a large number of algorithmic research papers on SNNs. It is for the moment however out of scope for this theoretical paper, which assumes that such a hardware may indeed be built. If desired, we will clarify this point in the updated version of our paper, and identify more clearly the assumptions we make on the hardware implementation.
>
> We are happy to reply to any further questions or remarks that you might have.

---

> ### Author Response · Authors · 2019-11-15
> **Paper update with more detailed discussion of hardware considerations**
>
> Based on your review, we updated the discussion section and added a more detailed treatment of the points you raised. We added references to current work on neuromorphic hardware and explain that hardware overheads have to be taken into account when assessing system level improvements in event-based SNN systems.

---

### Official Review · AnonReviewer1 · 2019-10-21
**Official Blind Review #1**

**Rating:** 6

**Review:**

EDIT After Rebuttal: My understanding of the contributions of this paper has improved. I now increase my score to a weak accept.

This paper proposes a new backpropagation algorithm learning algorithm "SpikeGrad" for Spike-based neural network paradigm. Simulating this algorithm on a classical hardware would require a lot of time-steps. To circumvent this, they show how to construct a corresponding artificial neural net (that can be trained using the traditional gradient based algorithms) which is equivalent to the spiking neural net. Using this equivalence they simulate a large scale SNN on many real-world dataset (first paper to do so). In particular, they use MNIST and CIFAR-10 for this purpose. They show that training a fixed architecture using their method is comparable to other prior work which uses high-precision gradients to train them. They also show how to exploit sparsity of the gradient in the back propagation for SNN.

This paper is hard-to-follow for someone not familiar with the background material. In particular, without looking at prior literature it was hard to understand that "integrate and fire neuron model" is essentially the feedforward mechanism for the SNN. I would suggest the authors make this a bit more explicit. Moreover, it would serve the structuring of the paper to have a formal "Preliminaries" section, where all known stuff goes. It was hard to discern what is new in this paper, and what is from prior work and these are mixed in section 2. For instance, section 2 states "SpikeGrad" algorithm; but the main contribution (ie., the back propagation algorithm) only appears in the middle of this section. Likewise, I think section 3 can be arranged better. In particular, the equivalence is a "formal" statement and thus, could be stated as a theorem followed by a proof. It will also make it explicit as to what does it mean by an "equivalent" network. In fact, it is still not clear to me at this point what that statement means. Could you please elaborate this in the rebuttal?

Regarding the conceptual contribution of this paper, if I understood things correctly, the main claim is that they give a new way to train SNN whose performance on MNIST and CIFAR-10 is comparable to other works. The second contribution is that they give the equivalence between ANN and SNN (point above). It is also unclear to me what the point regarding the sparse gradient in the backpropagation in the experimental section is trying to make? Could you please clarify this in the rebuttal as well?

At this point, the writing of this paper leaves me with many unanswered questions that needs to be addressed before I can make a more informed decision. Please provide those in the rebuttal and based on those will update my final score. But with my current understanding of this paper, I think this does not meet the bar. The contributions in this paper do not seem too significant.

**Experience Assessment:**

I do not know much about this area.

**Review Assessment: Checking Correctness Of Derivations And Theory:**

I assessed the sensibility of the derivations and theory.

**Review Assessment: Checking Correctness Of Experiments:**

I assessed the sensibility of the experiments.

**Review Assessment: Thoroughness In Paper Reading:**

I read the paper thoroughly.

---

> ### Author Response · Authors · 2019-11-08
> **Clarifications demanded by the reviewer**
>
> Thank you very much for your helpful review.
>
> As you have remarked correctly, the integrate-and-fire (IF) model is usually used for the forward propagation phase in an SNN. Our work uses the integrate-and-fire neuron model additionally to implement backpropagation, and is the first work to do so successfully on the CIFAR10 dataset using a large network.  The main interest in using spikes to implement backpropagation is that also training can be implemented "on-chip" in an event-based SNN hardware system that implements the IF neuron model (please note that SNNs are always implemented in dedicated, neuromorphic hardware, such as Loihi by Intel, that is optimized for the IF neuron model). SpikeGrad is therefore the only training method for SNNs so far that is able to perform spike-based training in a large network on the CIFAR10 dataset, and which yields performances comparable to SNN systems trained "off-chip" with floating point gradients.
>
> This also relates to your question why we present an analysis of the gradient sparsity. Because our method implements backpropagation with the IF model, it is able to exploit this sparsity when training is performed in a dedicated hardware implementation of the IF neuron. The main advantage of the IF neuron is that events and therefore computation is only triggered in proportion to the integrated activation (in the forward pass) or the integrated error (in the backward pass). The IF model is typically only analyzed in the context of forward propagation, but our results demonstrate that it could be indeed also very interesting for backpropagation, because the gradient becomes extremely sparse, and values become very low, as soon as the system response is approximately correct. This could be an interesting property for an embedded system with continuous learning, since fine tuning the network could be performed with very little events, and therefore little computation.
>
> This implementation of backpropagation with the IF model is not our only contribution. An important property of SpikeGrad is that both forward and backward propagation can be described by an equivalent ANN. This is the reason why also the forward processing pass is part of SpikeGrad. This has the advantage that we can simulate the behavior of the SNN efficiently on GPUs using the equivalent ANN, with the guarantee that the SNN that uses the IF model will give the same results. Usually simulation of SNNs uses clock-based simulation of the IF neuron model. This is however inefficient, since the event-based computation of the IF neuron model is not well suited to standard computing hardware, in particular GPUs. This is one of the main reasons why other training algorithms for SNNs cannot be tested on large networks, since researchers do often not have access to the necessary dedicated hardware (often ASICs that are produced in small numbers), and simulation on standard hardware becomes intractable. SpikeGrad offers a practical solution to this problem.
>
> As you requested, we provide a clarification of our notion of equivalence: when implementing the SNN using the SpikeGrad model described in the paper, the accumulated responses (the sum of all emitted spikes) of each IF neuron at the end of a propagation phase (forward and backward) will be equivalent to the corresponding integer values in the ANN. Since learning and inference is performed on these accumulated quantities in the SNN, the SNN learns and infers equivalently to the ANN.  An SNN using the same initial conditions and that is trained on the same data will therefore learn the same weights as the ANN and give the same inference responses. An example of such an equivalence can be found in our response to reviewer #2.
>
> We hope that we could clarify the points you raised in your review and we are happy to provide additional details if you have further questions. Your are right that the labeling of the sections might be not very clear. We will upload an updated version of the article that addresses these points as soon as we have obtained your feedback to our first response.

---

> > ### Comment · AnonReviewer1 · 2019-11-14
> > **Thanks**
> >
> > Thanks for your detailed discussion and response to my question. This has helped me better understand the contributions of your paper and the missing gaps in my understanding of the paper.

---

> > > ### Author Response · Authors · 2019-11-15
> > > **Thanks**
> > >
> > > We are happy that we could provide answers to your questions. We updated the paper to clarify points that you addressed.

---

### Official Review · AnonReviewer2 · 2019-10-23
**Official Blind Review #2**

**Rating:** 6

**Review:**

This paper shows how SNNs can integrate backpropagation with a second accumulation module that discretizes errors into spikes. In other words, the authors show how to translate event-based information propagation (used by SNNs) into a backpropagation method, which is the main contribution. The description of establishing the equivalence between an ANN and SNN in Section 3 is mostly well done. They perform empirical studies on MNIST and CIFAR-10 to demonstrate the effectiveness of SpikeGrad.


= Main Concerns =

1. It seems not clear to me why it is a good idea to introduce a second compartment with a threshold in each neuron as described in Eqn. 6.
2. I very much like the idea of "translating" an SNN into an ANN. I'm a bit confused about the computational complexity estimation of the SNN. In particular, it is not clear to me what is the practical implication of {n-n_min}/n_min < 0.035. Furthermore, in https://openreview.net/forum?id=rkg6PhNKDr, for ANNs on CIFAR-10, freezing 80% of the VGG19 parameters from the third epoch onwards only results in 0.24% drop in accuracy. I wonder if the advantage of SNN over ANN is still huge in this case.
3. I do not think experiments on MNIST are very useful, as the task is a toy task. I would suggest running at least one more experiment on CIFAR-100 or TinyImagenet.


**Experience Assessment:**

I do not know much about this area.

**Review Assessment: Checking Correctness Of Derivations And Theory:**

I did not assess the derivations or theory.

**Review Assessment: Checking Correctness Of Experiments:**

I assessed the sensibility of the experiments.

**Review Assessment: Thoroughness In Paper Reading:**

I read the paper at least twice and used my best judgement in assessing the paper.

---

> ### Author Response · Authors · 2019-11-08
> **Responses to your main concerns**
>
> Thank you very much for your helpful review.
>
> Regarding your concerns, here are our responses:
>
> 1. The second compartment is in principal necessary to discretize the gradient into spikes, in analogy to feedforward propagation. However, if the derivative of the activation function in equation (5) is calculated and saved before the backpropagation phase, the value of V is no longer required and the same compartment could be reused for U. This is however more of an implementation detail, and we used two compartments here to clarify that both forward and backward propagation use the integrate-and-fire model.
>
> 2. In our framework, the SNN is guaranteed to have the same accumulated response as the ANN. However, for each value, there is in principle a large number of combinations of +1 and -1 that lead to the same response. For instance the value 5 in the ANN can be represented in the SNN by 5 times +1 or by 10 times +1 and 5 times -1. In the first case there are only five spikes that have to be processed, in the second case there are 15. Since the SNN can always propagate additional spikes to correct its current response, the final sum of all spikes will always be equivalent to the output of the ANN.  How many spikes the SNN will use to encode a number depends on the order of the input spikes. If the input values are encoded in an irregular fashion, for instance the input value 5 is encoded by 5 negative spikes followed by 10 positive spikes, also the neurons in the network will first propagate spikes of one sign, and then spikes of another sign that may cancel each other out. This therefore produces useless computation. However, if the input is coded in a regular fashion, and 5 is coded by 5 positive spikes, also the neurons will respond more regularly and little corrective spikes are necessary. Empirically, we observe that for this regular input encoding, the number of spikes necessary to encode a number is close to the optimum, and on average there are only approximately 3.5% more spikes than the absolute value of the ANN activation (e.g. the value 100 in the ANN needs on average 103.5 spikes to be encoded in the SNN).
> You are correct that many ANNs are too large for the task they are used for, and possibly the number of neurons and parameters can be reduced, resulting in lower sparsity. However, SNNs are not only potentially more efficient in cases where a large number of values is exactly zero. It is sufficient that a large number of values is typically small. For instance, if 80% of activations in a 8-bit integer quantized ANN are 1, 15% are 2, and only the remaining 5% are much larger, almost all  values can be represented by only 1 or 2 spikes. This means these neurons will only trigger 1 or 2 accumulation operations in the receiving neurons (and no multiplications). On the other hand, a standard 8-bit integer ANN will perform a 8-bit MAC operation for each of these neuron outputs, regardless of their numeric value.
>
> 3. We compared our results to MNIST and CIFAR10 because these are still the most common benchmarks in SNN training. We will try to obtain results on CIFAR100 for the final version of the paper, we are however not sure if the time constraints will allow us to give reliable results by the end of the review period.
>
> We will add these points in the updated version of the paper as soon as we have obtained your feedback. We are happy to reply to any further questions or remarks that you might have.

---

> ### Author Response · Authors · 2019-11-15
> **Paper update and additional results on CIFAR100**
>
> Based on your review, we clarified the content in some sections and added additional results.
>
> As you requested, we added inference scores for the CIFAR100 dataset.  Using the same architecture as for CIFAR10, we obtain a maximal classification score of 64.40%. Running the same experiment with only the forward pass encoded in
> spikes, but floating point gradients, we obtain 64.69%. This result could probably be improved by
> adapting the architecture better to the dataset or adding additional regularization, but seems acceptable for a simple CNN without BatchNorm or residual connections. The error on the training set converges to around 98% over 300 epochs, so convergence of the optimization algorithm does seems to be a major problem. It therefore should demonstrate that coding the gradient into spikes does not lead to a large precision loss even in this scenario.
>
> We additionally extended the discussion section to treat better certain points that you raised regarding the efficiency of the spike coding scheme in ANNs with optimized parameter number.

---

### Decision · Program_Chairs · 2019-12-19

**Decision:**

Accept (Poster)

**Comment:**

This paper proposes a learning framework for spiking neural networks that exploits the sparsity of the gradient during backpropagation to reduce the computational cost of training. The method is evaluated against prior works that use full precision gradients and shown comparable performance. Overall, the contribution of the paper is solid, and after a constructive rebuttal cycle, all reviewers reached a consensus of weak accept. Therefore, I recommend accepting this submission.